# Immunogenic Properties of Recombinant Enzymes from *Bothrops ammodytoides* towards the Generation of Neutralizing Antibodies against Its Own Venom

**DOI:** 10.3390/toxins11120702

**Published:** 2019-12-02

**Authors:** Herlinda Clement, Ligia Luz Corrales-García, Damaris Bolaños, Gerardo Corzo, Elba Villegas

**Affiliations:** 1Centro de Investigación en Biotecnología, Universidad Autónoma del Estado de Morelos, Av. Universidad 2001, Cuernavaca Mor 62209, Mexico; linda@ibt.unam.mx; 2Departamento de Medicina Molecular y Bioprocesos, Instituto de Biotecnologia, Universidad Nacional Autónoma de México, Avenida Universidad, 2001, Apartado Postal 510-3, Cuernavaca Mor 62210, Mexico; ligia.corrales@udea.edu.co (L.L.C.-G.); damaris@ibt.unam.mx (D.B.); 3Departamento de Alimentos, Facultad de Ciencias Farmacéuticas y Alimentarias, Universidad de Antoquia, AA 1226, Medellín 050010, Colombia

**Keywords:** antibodies, *Bothrops ammodytoides*, metalloprotease, protein expression, serine-protease, snake, venom, viper

## Abstract

Bothropic venoms contain enzymes such as metalloproteases, serine-proteases, and phospholipases, which acting by themselves, or in synergism, are the cause of the envenomation symptoms and death. Here, two mRNA transcripts, one that codes for a metalloprotease and another for a serine-protease, were isolated from a *Bothrops ammodytoides* venom gland. The metalloprotease and serine-protease transcripts were cloned on a pCR^®^2.1-TOPO vector and consequently expressed in a recombinant way in *E. coli* (strains Origami and M15, respectively), using pQE30 vectors. The recombinant proteins were named rBamSP_1 and rBamMP_1, and they were formed by an N-terminal fusion protein of 16 amino acid residues, followed by the sequence of the mature proteins. After bacterial expression, each recombinant enzyme was recovered from inclusion bodies and treated with chaotropic agents. The experimental molecular masses for rBamSP_1 and rBamMP_1 agreed with their expected theoretical ones, and their secondary structure spectra obtained by circular dichroism were comparable to that of similar proteins. Additionally, equivalent mixtures of rBamSP_1, rBamMP_1 together with a previous reported recombinant phospholipase, rBamPLA2_1, were used to immunize rabbits to produce serum antibodies, which in turn recognized serine-proteases, metalloproteases and PLA2s from *B. ammodytoides* and other regional viper venoms. Finally, rabbit antibodies neutralized the 3LD50 of *B. ammodytoides* venom.

## 1. Introduction

Viper venoms are cocktails of toxic and non-toxic enzymes, as well as non-enzymatic proteins used both for constriction and ingestion of prey. Typical snake venom enzymes are L-amino acid oxidases, acetylcholinesterases, serine-proteases, phospholipases type A2, and metalloproteases [1]. *Bothrops ammodytoides*, a South American viperid from Argentina, has venom enzymes that produce dermonecrotic, inflammatory-edematogenic, and hemorrhagic effects in mice, and probably in humans [2]. Commercial antidotes or antivenoms against the bite of vipers contains antibodies that recognize such toxic enzymes, and neutralize them. Nowadays, antivenoms are generated by immunization of animals, regularly horses, using the whole venom of pit-vipers, having toxic or non-toxic components, in some cases being detrimental for such animals [3]. The immunogenicity of serine-proteases, metalloproteases, and phospholipases type A2, was assessed. Thus, the venom and glands of *B. ammodytoides* were used as a model to obtain such enzymes and then express them in a recombinant way. The aim was to develop neutralizing antibodies without the immunization of animals with the whole viper venom. The heterologous expression of such enzymes could support scientific research concerning the properties of venom toxic enzymes and, eventually, the enrichment of venom immunogens for the production of viperid antivenoms. Therefore, in this study, we report the cDNA cloning and recombinant production of a serine-protease (rBamSP_1) and a metalloprotease (rBamMP_1) from the venomous gland of *B. ammodytoides*. Furthermore, the recombinant serine-protease and metalloprotease, together with a previous reported recombinant phospholipase A2, from the same viperid species, were used as immunogens to generate rabbit antibodies, which reduced the enzymatic activities of proteases and phospholipases as well as neutralized the toxicity of *B. ammodytoides* venom. Therefore, here we communicate the proof of concept that a mixture of recombinant viper enzymes, specifically serine-proteases, metalloproteases, and phospholipases type A2, could be used as immunogens, instead of the whole venom, to generate neutralizing antibodies against pit-viper venoms, and such antibodies could be useful for bothropic antivenom studies.

## 2. Results and Discussion

### 2.1. Isolation and Sequence Determination of rBamSP_1 and rBamMP_1

The N-terminal sequence of the two enzymes was the starting point to design a set of synthetic primers used to amplify the sequences of serine-protease and metalloprotease from *B. ammodytoides* (see Materials and Methods). Applying polymerase chain reaction (PCR), we amplified both enzymes starting with the previously obtained cDNA. The purified inserts were cloned into plasmids pCR^®^2.1-TOPO^®^ and then into the plasmid pQE30 (Appendix A). The sequence of amino acid of the cloned rBamSP_1 and rBamMP_1 are shown (Table 1 and Table 2). rBamSP_1 preserves the highly conserved residues His56, Asp100 and Ser197, of the common serine-protease active site (Table 1, bold); apparently, a characteristic of snake venom-type thrombin-like enzyme and rBamMP_1 has a consensus zinc-binding motif HEXGHXXGXXHD (Table 2, bold) [4].

### 2.2. Expression of rBamSP_1 or rBamMP_1, Their Purification and Folding

The genes of rBamSP_1 or rBamMP_1 were cloned in pQE30 plasmids, respectively, which generates proteins with a 6His-tag attached to the N-terminal region, allowing a fast purification of the recombinant proteins by nickel affinity agarose columns (NiNTA). We also added a recognition site for the peptidase FXa, located between the 6His-tag and the enzyme sequence of rBamSP_1 or rBamMP_1, in case of potentially harmful influence of the tag toward the biological activity. Heterologous expression of either rBamSP_1 or rBamMP_1 was achieved using *E. coli* Origami and M15 strains, respectively (Appendix A). rBamSP_1 or rBamMP_1 were predominantly located in inclusion bodies (Appendix A, lane 2 for rBamSP_1, and Appendix A lane 4 for rBamMP_1), and were recovered by NiNTA (Appendix A, lanes 7-13 for rBamSP_1 and Appendix A lanes 8-14 for rBamMP_1). The heterologous expression of either rBamSP_1 or rBamMP_1 from inclusion bodies, as well from NiNTA purification, was verified through Western blot, utilizing an anti-6His-tag antibody, joined to alkaline phosphatase. Either rBamSP_1 or rBamMP_1 was subjected to in vitro folding and then a reversed-phase high-performance liquid chromatography (RP-HPLC) purification. All the fractions with retention times fluctuating from 34 to 40 min (linear-gradient, 10%–60% of B in 50 min) were gathered (Figure 1). Sodium dodecyl sulfate polyacrylamide gel electrophoresis (SDS-PAGE, 12%) confirmed that the collected fractions possessed the expected molecular mass. Furthermore, either rBamSP_1 or rBamMP_1 showed the experimental molecular masses 27,849.3 and 26,830.2 (Appendix A), which were acquired by mass spectrometry and corresponded to the predicted molecular masses of enzymes 6His-tagged. The results indicated that the fractions collected from the RP-HPLC purification corresponded to rBamSP_1 or rBamMP_1; although, the enzymes contain 12 and 10 cysteines, and they could probably produce up to 10,395 and 945 different cysteine linked isoforms, respectively, regarding combining disulfide pairing probabilities. Evidently, the cellular system in the snake venom gland dictates the establishment of accurate protein folding of native enzymes when compared to a heterologous expression system. The protein yield of rBamSP_1 and rBamMP_1, were 1 and 1.8 mg/L, respectively.

### 2.3. Secondary Structure of rBamSP_1 and rBamMP_1 

Circular dichroism (CD) was used to analyze the recombinant serine- and metalloprotease; the aim was to confront their secondary structures upon those already reported. The results showed significant absorption for α-helix and β-strand content (secondary structures) (Figure 2). In agreement with the CD deconvolution software [5], the secondary structure was 35.2%, 12.6%, 7.5%, 44.7% and 20.7%, 26.4%, 14.0% and 38.9% of α-helix, β-strands, turns and random coil structure for rBamSP_1 and rBamMP_1, respectively. Until now, all pit-viper venom serine- and metalloproteases comprise a higher balance of α-helix, which is described by a positive band (198 nm) and a negative ellipticity (208–222 nm) [6]. Also, they contain a higher proportion of β-strands. Here, rBamSP_1 and rBamMP_1 showed similar patterns to the *Bothrops jararacussu* serine protease (PDB: 4GSO) and *Daboia siamensis* metalloprotease (PDB: 2E3X), respectively. As reported earlier, the recombinant rBamPLA_2__1 shows also secondary structure similar to the native pit-viper phospholipases [7]. Overall, the three recombinant proteins agreed with the canonical secondary structures of phospholipases and proteases from viper venom [6,7].

### 2.4. Rabbit Immunization, Antibody Recognition and Titers

Rabbits were immunized with 10 mg of a blend of recombinant enzymes rBamSP_1, rBamMP_1, and rBamPLA_2__1. The rabbits were bled after 5 months of immunization, and the serum was evaluated in its capacity to recognize each recombinant enzyme, and also the native enzymes from Bothrops venoms. Figure 3A presents the venoms protein pattern from different species of Bothrops, where serine-proteases, metalloproteases, and phospholipases are evident from their apparent molecular masses (red triangle marks in Figure 3A,B). Figure 3B illustrates how rabbit antibodies recognize proteins of venom from different *Bothrops* species. We perceived antibody-recognition for a wide span of proteins, which represents the native serine-proteases, metalloproteases, and PLA2s from *Bothrops* venoms. Furthermore, the same rabbit serum antibodies were used to titer the antibody recognition against several bothropic venoms, Figure 4 presents the rabbit antibody titers toward the recombinant enzymes and the venom of other *Bothrops* species. Table 3 shows that the recombinant enzymes were the most recognized as expected, and among them, the recombinant phospholipase had the higher titers followed by that of the serine- and metalloproteases. Concerning recognition of rabbit IgGs to pit-viper venoms such *B. ammodytoides, B. diporus, B. jararacussu, B. moojeni, B. alternatus,* and *B. asper,* the venoms of *B. ammodytoides* and *B. moojeni* were the most recognized with titer values of 543 and 624, respectively (Table 3, see Section 4.9. Enzyme-Linked Immuno Sorbent Assay (ELISA) in the Material and Methods). However, the antibody recognition was very evident; the EC50 (half-maximal effective concentration) was considerably low for *B. asper* and B. *diporus* (85 and 90, respectively. Table 3). Nevertheless, the rabbit antibodies were capable of reducing the enzymatic activities of proteases and phospholipases from *B. ammodytoides* venom. Figure 5 shows the phospholipase activity of *B. ammodytoides* venom in the absence and presence of IgGs from immunized rabbits. Likewise, Figure 6 shows the *B. ammodytoides* venom protease activity in a gelatine zymogram in the absence and presence of IgGs from immunized rabbits. This data proves that rBamSP_1, rBamMP_1, and rBamPLA2_1 could be used as immunogens for producing rabbit antibodies to decrease phospholipase and protease activities, which are related to pit-viper venom poisonous activities, such as myotoxicity or restraint to platelet function that conducts to death, at least with the venom of *B. ammodytoides*.

The reactivity of the antibodies anti-rBamSP_1, rBamMP_1, and rBamPLA2_1 over other *Bothrops* venoms may also heighten the interest in the research on the potential neutralization of harmful activities by recombinant enzymes as immunogens. Currently, the production of anti-snake venom for healing objectives are obtained in form of polyclonal antibodies generated in large vertebrates, often applying whole venom from a viper as immunogen; other strategies are being analyzed [8], for example, the heterologous proteins obtained through molecular biology techniques, could be used for the production of polyclonal or monoclonal antibodies or neutralizing fragments, which may indicate future and interesting approaches for anti-venom production [9,10].

### 2.5. Immunoaffinity Chromatography and Venom Neutralization

Intended to deep into antibody recognition and venom neutralization, immunoaffinity chromatography was performed, which is generally used in antivenomics for the analysis of the quality of antibodies when bound to animal venoms [11]. Figure 7A shows the HPLC profile of the *B. ammodytoides* venom. Here, the immunoaffinity chromatography (Figure 7B,C) using a ligand density (mg IgG/mL of the matrix) significantly showed the union of venom enzymes to the rabbit antibodies, which recognized metalloproteases, serine-proteases and phospholipases from *B. ammodytoides* venom (inside, Figure 7C, Appendix A). The information obtained here could be valuable for obtaining a better comprehension of venom protein immunogenicity, antivenom strength, and for improving antivenom quality.

The practical estimation of antivenoms is established upon its capacity to neutralize the harmful impact of venoms (LD_50_). Consequently, to observe venom neutralization by the rabbit IgGs, the LD_50_ for *B. ammodytoides* venom was determined, which was 18.6 µg of venom/mouse (0.93 mg/kg mouse). Considering the interspecific variation in the composition of the snake venom, the previous value is within the LD_50_s reported from *Bothrops* venom, which are ranging from 0.66 to 232 mg/kg mouse [12,13,14,15]. Afterwards, the rabbit anti-recombinant enzyme IgGs were challenged with 3LD50 (55.8 µg/mouse) of *B. ammodytoides* venom, according to the World Health Organization (WHO) [16]. The IgGs from the anti-recombinant enzymes neutralized the lethality of *B. ammodytoides* venom with an ED_50_ of 13.1 mg IgGs/mouse (n = 3), meaning 55.8 µg of whole *B. ammodytoides* venom neutralized by 13.1 mg of anti-recombinant enzymes IgGs; that is, an EaV value of 4.3 µg of venom/mg of IgGs (233 mg of IgGs/mg of venom). Although few communications had reported the specific activity of IgG anti-venoms, some specific activity values from commercial antivenoms (obtained from horse IgGs) are from 2.8 to 26.8 mgAv/mg *B. asper* venom) [12,15].

## 3. Conclusions

This work presents the proof of the concept of the use of a mixture of recombinant enzymes (metalloprotease, serine-protease, and phospholipase) to produce antibodies against native venom enzymes from species of *Bothrops.* The circular dichroism spectra of the venom enzymes here heterologously expressed indicate that the secondary structure of such recombinant enzymes is maintained, even though some isoforms having different disulfide pairings could take place, and such structures could produce antibodies against native enzymes. Such recombinant proteins are immunogenic, they yield neutralizing antibodies against *B. ammodytoides* venom, and such antibodies recognize related enzymes from other *Bothrops* venoms. Neutralization of whole animal venoms or venom components such as elapids (*Micrurus, Dendroapsis, Naja*), scorpions (*Centruroides, Tityus, Androctonus*) and spiders (*Loxoceles*), using antibodies raised up by recombinant proteins such as immunogens, has been already reported [7,8,9,10]. Likewise, recombinant enzymes from viperids could be employed to produce antibodies to neutralize the toxicity of envenomation, such as platelet alterations and myotoxicity, both generated by venom from related *Bothrops* vipers. In fact, since the amino acid domains into metalloproteases, serine-proteases, and phospholipases are mostly conserved in other bothropic venoms, here we postulate that this experimental antivenom could neutralize or retard the venom activity of such *Bothrops* species.

## 4. Materials and Methods

### 4.1. Venom Gland and Venom

Adult exemplars of *B. ammodytoides* were maintained in healthy conditions and clean boxes, at 27 °C (constant temperature) with cycles of 12 h of light and dark. Every fortnight the animals were fed, and water was administered ad libitum. The venom was manually collected, and at once, dried under vacuum and saved at –20 °C for further use. A healthy *B. ammodytoides* snake was selected to obtain a single venom gland (one or two). The animal was anesthetized, and by chirurgical extraction, the gland was removed and treated instantaneously with RNAlater^®^ (Thermofisher, Asheville, NC, USA) and saved at –70 °C till use. Following the chirurgical procedure, the snake was recovered and remained healthy.

### 4.2. Bacterial Strains, Enzymes and Plasmids

For DNA cloning and plasmid proliferation, we used the XL1-Blue *Escherichia coli* strain. *E. coli* strains, Origami or M15, were used for the expression of recombinant serine-protease and metalloprotease, respectively. Plasmids pCR^®^2.1-TOPO^®^ (Invitrogen, Carlsbad, CA, USA), and pQE30 (Qiagen, Valencia, CA, USA) were employed for cloning the genes and the expression of the 6His-tagged recombinant proteins, respectively. Factor Xa protease (FXa), restriction enzymes, T4 DNA ligase, and Taq polymerase, were acquired from New England Biolabs (NEB, Ipswich, MA, USA). 

### 4.3. RNA Extraction and Gene Assembly

“Total RNA Isolation System” (Qiagen, Valencia, CA, USA) was used to extract the total RNA from the venom gland. The N-terminal sequence of enzymes, which allowed us to design specific oligonucleotides, and all of them used to amplify (by 3’RACE) a cDNA of 707 bp and 662 bp of rBamSP_1 and rBamMP_1, respectively [17]. The oligonucleotides were labeled as Oligo1 Fw serine-protease (GTCATTGGAGGTGATGAATGT, Tm 60 °C), Oligo AUAP from 3’RACE kit (GGCCACGCGTCGACTAGTAC), Oligo 1 Fw metalloprotease (GAGCCCATCAAAAAGGCC, Tm 56 ºC) and Oligo 1 Rv metalloprotease (GGCATCGAAGCGATTTCT, Tm 58 °C).

After each construction and transformation, some clones were picked and analyzed upon the molecular biology protocols. The recombinant gene was confirmed by plasmid sequencing (plasmid purification by the High Pure Plasmid Isolation Kit, Roche, Basel, Switzerland).

### 4.4. Plasmid Construction for Protein Expression

The genes of enzymes rBamSP_1 and rBamMP_1 also included the recognition sequences for both restriction enzymes (*Bam*HI and *Pst*I) and protease FXa. The digested inserts were ligated into the expression vector pQE30 using the same restriction sites, *Bam*HI and *Pst*I. The pQE30 vector contains a tag of polyhistidine (6His) to assist in protein purification by affinity chromatography. Between the 6His-tag and the toxin was placed the FXa cleavage site, intended to obtain the entire native protein if necessary (Appendix A). The constructions in pQE30 were confirmed by DNA sequencing. Competent *E. coli* Origami and M15 cells were transformed with the vector corresponding to each enzyme by the following procedure: incubation for 30 min on ice, heat-shocked at 42 ºC for 1 min, and finally, incubation in ice for 5 min. The transformed cells were grown at 37 ºC for one h in super optimal broth with catabolite repression (SOC) medium and then poured on Luria Broth (LB) agar, including 100 µg/mL of ampicillin.

### 4.5. Expression and Purification of the Enzymes rBamSP_1 and rBamMP_1

The *E. coli* Origami strain harboring the plasmid pQE30rBamSP_1 or *E. coli* M15 strain transformed with the plasmid pQE30rBamMP_1 were cultivated in LB liquid medium. Once a given bacterial culture reached an optical density (OD600) of 0.8 absorption units, it was supplemented with 0.5 mM isopropyl-ß-D-thiogalactopyranoside (IPTG) and incubated at 16 ºC for 24 h. Afterward, the culture was centrifuged (7168× *g* for 20 min in JA-14 rotor, Beckman model J2-21). Cells were resuspended in washing buffer (Tris-HCl 0.05 M, pH 8.0) and disrupted using the protein extraction reagent BugBuster^®^ (Novagen, Germany). The resulting cell debris were centrifuged (16,128× *g* for 20 min), the soluble fraction was discarded, and the insoluble fraction was reused.

The insoluble fraction was cleaned doubly using the washing buffer and then centrifuged at 16,128× *g* for 20 min. The insoluble fraction contained the inclusion bodies, which were resuspended and solubilized in guanidinium chloride solution (GndHCl 6M, Tris-HCl 0.05 M, pH 8.0). The solution was centrifuged at 16,128× *g* for 20 min; after centrifugation, the insoluble material was removed. The supernatant solution, which contained the recombinant protein, was purified by affinity column chromatography that included a Ni-NTA (Ni-nitrilotriacetic acid) resin. The system was equilibrated with buffer A’ (GndHCl 6M, Tris-HCl 0.05M, pH 8.0), and the recombinant proteins were eluted with buffer B’ (GndHCl 6M, Tris-HCl 0.05M, imidazole 400 mM, pH 8.0). The eluted samples were subjected to the second purification step, reversed-phase HPLC (RP-HPLC) using an analytical C4 reversed-phase column (Vydac 214 TP 4.6x250 mm, Hesperia, CA, USA), and applying a linear gradient with solvent A (0.1% trifluoroacetic acid, TFA, in water) and solvent B (0.1% TFA in acetonitrile). The linear gradient was run from 10 to 60% solvent B for 50 min at 1 mL/min. Eluted proteins, detected at 230 nm, were collected, and immediately lyophilized.

### 4.6. Molecular Mass Determination

The molecular masses of the enzymes were validated by mass spectrometry analysis. Fractions of rBamSP_1, collected from RP-HPLC, were lyophilized, and then, concentrated to 500 pmol/5 μL with a solution of acetonitrile 50% and acetic acid 1%. Samples were directly injected into a Thermo Scientific LCQ Fleet ion trap mass spectrometer (San Jose, CA, USA) with a Surveyor MS syringe pump delivery system. The samples were run at 10 μL/min, and they were split to introduce only 5% of the sample into the nanospray source (0.5 μL/min). The spray voltage and capillary temperature were fixed at 1.5 kV and 150 ºC, respectively. The fragmentation source was operated at 25–35 V of collision energy, 35–45% (arbitrary units) of normalized collision energy, and all spectra were gathered in the positive-ion mode. Data acquisition and deconvolution were made on the Xcalibur Windows NT PC data system. For rBamMP_1, samples were desalted with Zip Tip C4 (Millipore; Billerica, MA, USA), and injected to an LC–MS (liquid chromatography–mass spectrometry) system composed of an Accela Pump (Thermo-Fisher Co, San Jose, CA, USA) coupled to a LTQ-Orbitrap Velos mass spectrometer (Thermo-Fisher Co., San Jose, CA, USA) with a nano-electrospray (ESI) ionization source. The nano-flow liquid chromatography consisted of an isocratic system of 50%–50% of solvents A (water) and B (acetonitrile), both with 0.1% formic acid with a running time of 20 min. A homemade hair needle (ID 0.75 µm and 20 cm long) was used, and the flow of the LC system was 300 nL/min.

### 4.7. Circular Dichroism

Secondary structure spectra of rBamSP_1 and rBamMP_1 were registered at room temperature in quartz cells (1 mm-path) from 190 to 260 nm employing a spectropolarimeter Jasco J-710 (Jasco, Japan). Individually protein was dissolved in trifluoroethanol (60%) to obtain a concentration of 0.6 mg/mL. The delta CD absorbances were collected every 1 nm at a speed of 20 nm/min. The CD values corresponded to the average of three different CD registrations, and they were examined using the algorithms presented online at the BeStSel (Beta Structure Selection) [5].

### 4.8. Animal Immunizations

Rabbits were hyperimmunized by the subcutaneous way, with 10 mg of a blend of the three recombinant enzymes rBamSP_1, rBamMP_1, and rBamPLA2_1, 10/3 mg each, to produce serum antibodies. The immunization protocols started with the administration of a dose of 0.1 mg of the mixed proteins into the Complete Freud’s Adjuvant (CFA); afterward, increasing doses up to 0.25 mg, and shifting with incomplete Freud’s (IFA) and aluminum hydroxide (AH), were administered during one month. Immunizations were made every two weeks, with 0.5 mg of the mixed proteins. The serum from rabbits (n = 3) was pooled, and antibodies were obtained from the plasma through caprylic acid precipitation (5%) [18]. A final solution of rabbit-derived immunoglobulins containing 50 mg/mL of protein was used for further experiments.

### 4.9. Enzyme-Linked Immunosorbent Assay (ELISA)

Enzyme-linked immunosorbent assays (ELISA) were run, according to Clement et al. [7]. Briefly, solid-phase-adsorbed rBamSP_1, rBamMP_1 or rBamPLA2_1 were prepared by treating wells of MaxiSorp plates (NUNC™, Thermo Scientific, Waltham, MA, USA) with 100 µL of a solution of the recombinant protein (5 µg/mL) in 0.1 M sodium carbonate buffer (pH 9.6). After incubation at 4 °C overnight, wells were emptied and washed three times with 200 µL of washing buffer (Tris-HCl 0.05 M, pH 8, Tween-20 0.5 mg/mL, and NaCl 0.15 M). Then, the wells were loaded with 200 µL of blocking buffer (gelatin 5 mg/mL, Tween-20 2 mg/mL, and Tris-HCl 0.05 M pH 8). After 2 h incubation at 37 °C, wells were emptied and washed as explained before and filled again with aliquots of 100 µL each, which contained rabbit IgGs anti- rBamSP_1, rBamMP_1, and rBamPLA2_1, serially diluted and prepared in incubation buffer (Tris-HC 0.05 M, pH 8, gelatin 1 mg/mL and NaCl 0.5 M). The first dilution was 1:30, and the incubation time was 1 h at 37 °C. After washing, the bound rabbit IgGs were allowed to react with 100 µL per well of 0.1 mg/mL of anti-rabbit IgGs, labeled with horseradish peroxidase (Merck KGaA, Darmstadt, Germany), prepared in incubation buffer. After one hour of incubation at 37 °C, wells were emptied, washed, and refilled with 100 µL of the solution ABTS (Roche, Basel, Switzerland), which was the peroxidase substrate. The reaction of color development was halted by the addition of 25 µL of SDS (20%), and the samples were read at 405 nm in a Microplate Reader (Tecan Sunrise IVD version, Tecan Trading AG, Switzerland). Data were analyzed by nonlinear regression using the sigmoidal dose-response equation from the Prism program (Graphpad Prism v. 6.0c, San Diego, CA, USA). Conventional titers were calculated from the midpoint of the curve and correspond to the IgGs dilution for half of the maximal binding, which was considered as the half-maximal effective concentration (EC50).

### 4.10. Phospholipase Activity

The phospholipase A2 activity in the venom *B. ammodytoides* was measured in the absence and presence of rabbit serum immunoglobulins hyperimmunized with the mixture of the three recombinant enzymes, rBamSP_1, rBamMP_1, and rBamPLA_2_. It was indirectly monitored by titration of fatty acids with NaOH as a result of the hydrolysis of egg yolk phospholipids caused by the *Bothrops* venom. The activity was represented as µmol of NaOH/min*mg of the enzyme, as describe by Shiloah et al. [19].

### 4.11. Protease Activity Using Gelatine Zymography

Briefly, the electrophoresis system described by Laemmli (1970) [20] was modified only by the presence of gelatin, which was co-polymerized within the separating gel in a concentration of 1.5 mg/mL. After electrophoresis, the gel was incubated with 50 mL of the buffer solution of Tris-HCl 0.1 M, Triton X-100 5%, pH 8, for one hour. Afterward, the gel was rewashed with the same buffer but including only Triton X-100 at 0.05% and incubated for another hour. Finally, the gel was incubated in the buffer without Triton X-100 for one more hour. Later, the gel was incubated inside a humid container overnight at room temperature, and then dyed with Coomassie brilliant blue G 250 for one hour. At last, it was distained with acetic acid 10% plus isopropanol 10%. The whole venom of *Bothrops* asper was used as a positive control.

### 4.12. Immunoaffinity Chromatography

The previous rabbit-derived immunoglobulins containing 50 mg/mL were used to prepare affinity chromatography columns with Sepharose 4B resin activated with cyanogen bromide (Sigma-Aldrich, Ontario, Canada) according to Pla et al. (2012) [21]. Once the venom was eluted from the affinity chromatography column, it was fractioned using an analytic C8 RP-HPLC column (5C8MS, 4.6 × 250 mm, Zorbax Agilent) using a linear gradient with solvent A (trifluoroacetic acid (TFA) 0.1%, in water) and solvent B (TFA 0.1% in acetonitrile). The linear gradient was conducted from 0 to 60% solvent B, for 60 min at 1 mL/min, fractions were detected at 280 nm, collected and immediately lyophilized. For protein identification, the samples were previously reduced with dithiothreitol (DTT, Sigma-Aldrich; St Louis, MO, USA), alkylated with iodoacetamide (Sigma-Aldrich) and digested "in solution" with Trypsin (Promega Sequencing Grade Modified Trypsin; Madison, WI, USA). The polypeptides generated by the enzymatic cleavage were desalted with Zip-Tip C18 (Millipore; Billerica, MA, USA) and injected to an LC-MS system composed of an EASY-nLC II nanoflow pump (Thermo-Fisher Co; San Jose, CA, USA) coupled to a LTQ-Orbitrap Velos MS (Thermo-Fisher Co, San Jose, CA, USA) with a nano-electrospray ionization (ESI) source. The total ion scanning (full scan) was performed on the Orbitrap analyzer with a resolution power of mass (RP Power; RP = m/FWHM) of 60,000. Peptide fragmentation was performed using the methods of CID (collision-induced dissociation) and HCD (high-energy collision dissociation). All spectra were acquired in positive detection mode. The execution and capture of the fragmentation data were performed depending on the total ion scan according to the pre-determined charges (only ions with a z2+, z3+, and z4+ charge were fragmented) with an isolation width of 2.0 (m/z), normalized collision energy of 35 arbitrary units, Q activation of 0.250, activation time of 10 ms and maximum injection time of 10 ms per micro-scan. Protein identification was performed with the spectrometric data in raw format in the Proteome Discoverer program 1.4.1.14 (Thermo-Fisher Co., San Jose, CA, USA) through the Sequest HT search engine. For the identity search, the bothrops.fasta (UniProt) protein database was used. A false discovery rate (FDR, minimum) of 0.01 and 0.05 (maximum) was applied in addition to the inverted database (Decoy database) as a tool of the "Percolator" validation program. The maximum tolerance of molecular mass difference of the precursor ion, when compared to the theoretical versus experimental values (precursor mass tolerance), was 20 ppm and the tolerance for the fragments obtained by dissociation of the precursor ion (fragment mass tolerance) was 0.6 Da. For the automatic search, the following modifications were established: carbamidomethylation constants of cysteines (C) and variables, oxidation of methionines (M), and deamination of asparagine (N) and glutamine (Q).

### 4.13. Protecting Activity of Immunoglobulins

Neutralizing activity of antibodies anti- rBamSP_1, rBamMP_1, and rBamPLA2_1 in vivo was developed according to the guidelines of our Institute Committee of Animal Welfare using the mouse as an animal model. For neutralization experiments, 3LD_50_ of whole *B. ammodytoides* venom was pre-incubated 30 min at 37 ºC with absence or presence of the rabbit antibodies (6.4, 9.2, and 13 mg/mL). Then groups (n = 3) of male mice (CD-1, 18–20 g body weight) were injected by the intravenous route [16]. After 24 h, mice conditions were observed.

### 4.14. Statistics

For statistics, the software Prism 4.0 (Graph Pad Inc., San Diego, CA, USA) was used. Results were expressed as mean and standard deviation, or as mean with 95% confidence intervals.

## Figures and Tables

**Figure 1 toxins-11-00702-f001:**
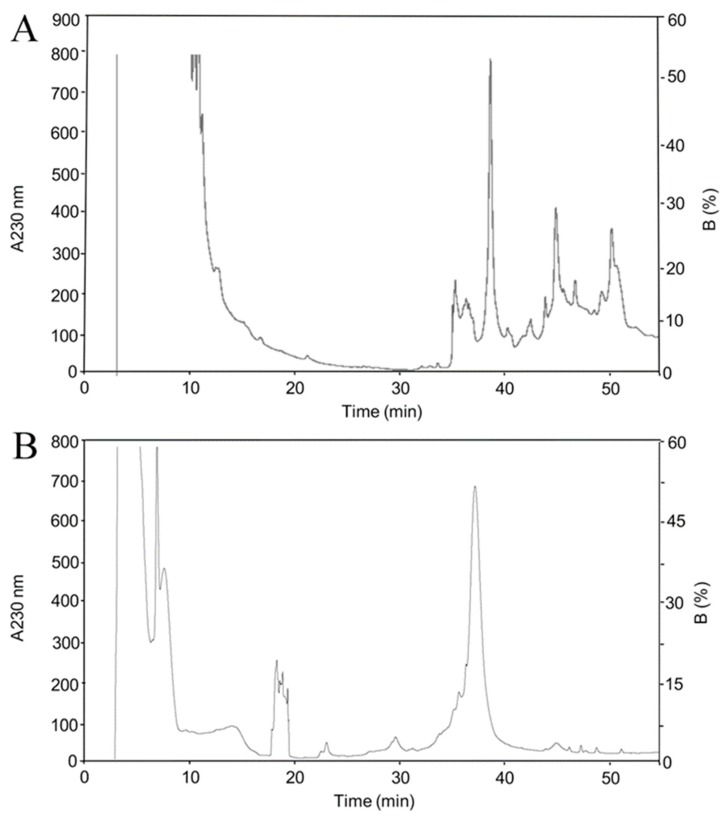
Chromatographic separation after Ni purification. (**A**) rBamSP_1; (**B**) rBamMP_1. The high-performance liquid chromatographic (HPLC) separations of either rBamSP_1 or rBamMP_1, obtained after elution from affinity chromatography (400 mM imidazole). The recombinant proteins were fractionated using an analytic C4 column (Vydac 214 TP 4.6 × 250mm, reverse-phase, USA) with solvent A (0.1% trifluoroacetic acid, TFA, in water), and solvent B (0.1% TFA in acetonitrile). From 10 to 60% of the B-solvent was run for 50 min at 1 mL/min, the fractions were detected at 230 nm. The different fractions were examined by mass spectrometry.

**Figure 2 toxins-11-00702-f002:**
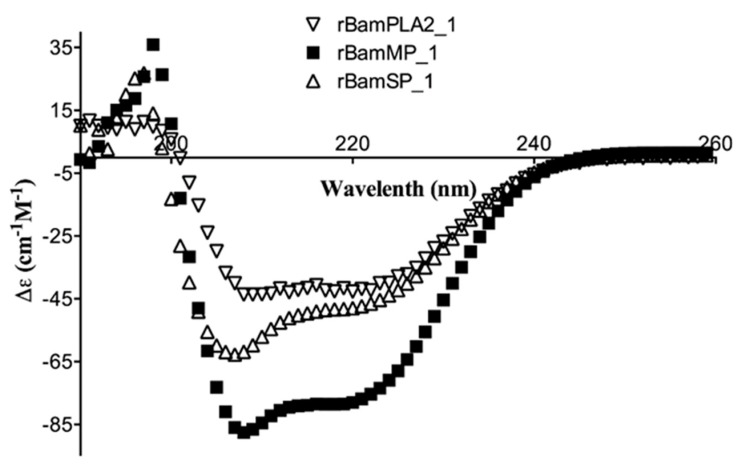
Circular dichroism (CD) of recombinant proteins rBamSP_1, rBamMP_1 and rBamPLA_2__1.

**Figure 3 toxins-11-00702-f003:**
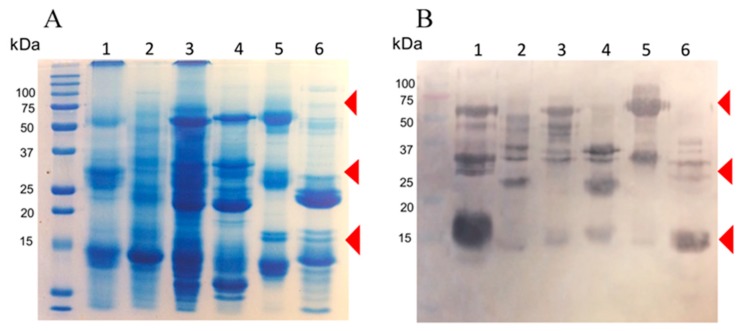
The proteins present in venoms samples from various species of *Bothrops* exposed in (**A**). sodium dodecyl sulfate (SDS) gel and (**B**). Western blot. Lanes, (1) *B. ammodytoides*; (2) *B. jararacussu*; (3) *B. diporus*; (4) *B. moojeni*; (5) *B. alternatus*; (6) *B. asper*. Each line of the SDS gel had 50 µg, and 10 µg in each line of the Western-blot (rabbit IgG anti-mixture of recombinant enzymes (rBamSP_1, rBamMP_1, and rBamPLA2_1) was the first antibody, and rabbit IgG coupled to alkaline phosphatase was the second antibody).

**Figure 4 toxins-11-00702-f004:**
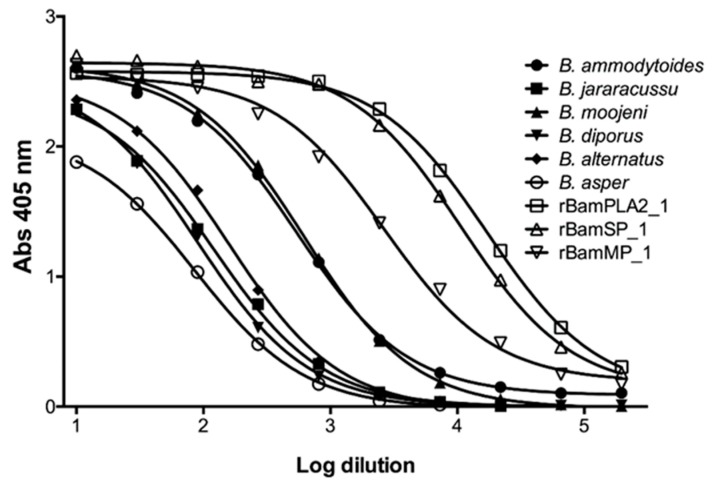
Antibody recognition of rabbit IgGs towards venoms of different *Bothrops* species. Titers of rabbit anti-recombinant enzymes (rBamSP_1, rBamMP_1, and rBamPLA2_1) against various venoms of *Bothrops* species, and also towards the recombinant enzymes used as immunogens.

**Figure 5 toxins-11-00702-f005:**
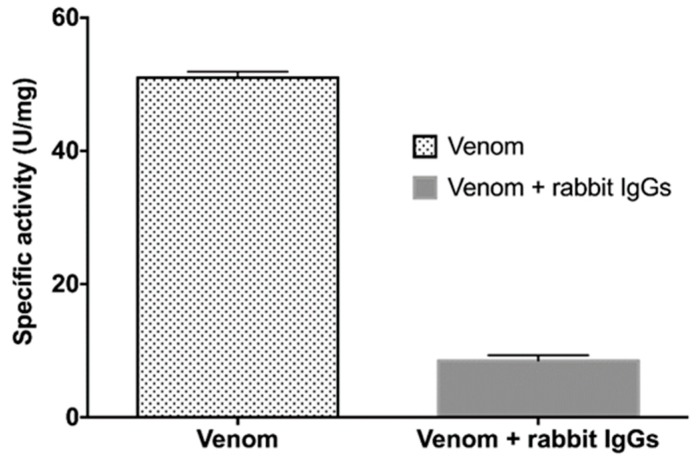
Phospholipase activity in the presence of rabbit IgGs. Titrimetric assays of PLA_2_ activity of *B. ammodytoides* venom, and inhibition of the same venom with the rabbit IgGs obtained from the immunization with rBamSP_1, rBamMP_1 and rBamPLA_2__1. The specific activity is given in U/mg = μmol of NaOH consumed per minute per milligram of venom. Error bars express the standard deviation of three experiments.

**Figure 6 toxins-11-00702-f006:**
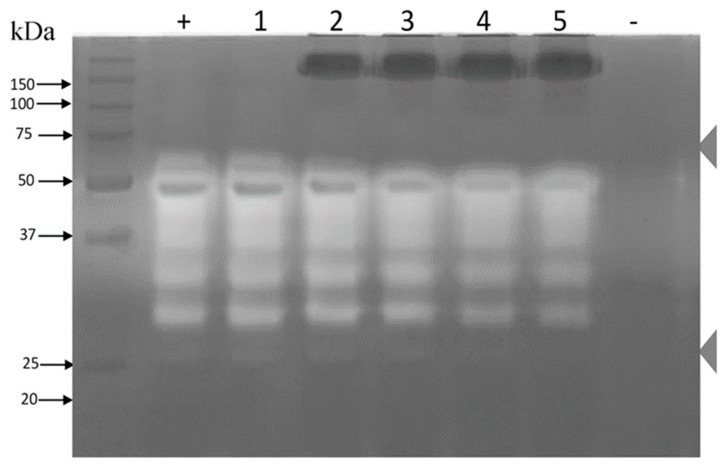
Zymogram of protease inhibition in the presence of rabbit IgGs. A 12% sodium dodecyl sulfate polyacrylamide gel electrophoresis (SDS-PAGE) gel with 1.5 mg/mL gelatin was generated. The whole venom of *B. ammodytoides* was used at 15 μg/well in all loading wells. Lanes (+) and (1) venom; (2) venom plus 50 μg of rabbit IgG; (3) venom plus 100 μg of rabbit IgG; (4) venom plus 150 μg of IgG; and (5) venom plus 200 μg of rabbit IgG; (-) Control of phosphate-buffered saline (PBS) 1×. The zymogram was incubated overnight at room temperature.

**Figure 7 toxins-11-00702-f007:**
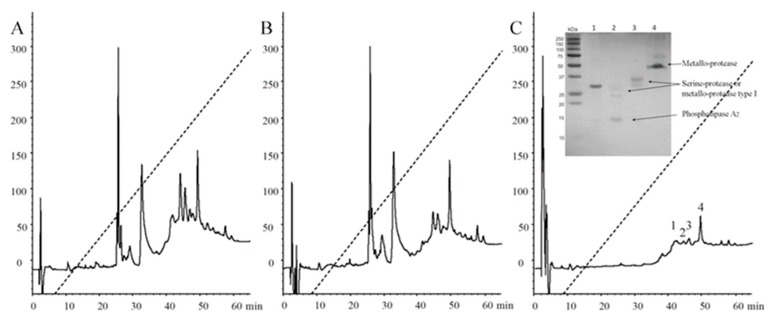
Reversed-phase high-performance liquid chromatography (RP-HPLC) profiles of *Bothrops ammodytoides* venom and protein fractions after immunoaffinity chromatography. (**A**) Complete venom of *Bothrops ammodytoides*; (**B**) Venom no bound to immunoaffinity column; and (**C**) Venom bound. The complete venom and the proteins eluted after immunoaffinity chromatography were fractioned by using an analytic C8 column (reverse-phase) with solvent (**A**) (0.1% trifluoroacetic acid, TFA, in water), and solvent (**B**) (0.1% TFA in acetonitrile). From 0 to 60% of the B-solvent was run for 60 min at 1 mL/min, the fractions were detected at 280 nm. For protein identification of caption in (**C)**, please see Section 4.12. Immunoaffinity chromatography.

**Table 1 toxins-11-00702-t001:** Amino acids residues of rBamSP_1 and alignment to similar viper enzymes.

rBamSP_1	Sequence ^1^
	56
rBamSP_1	VIGGDE*C*NINEHPFLVALYTSRSRRFH*C*GGTLINQEWVLTAAH*C*DRKNIRIKLGMHSKN
Q7T229.1	IIGGDE*C*NINEHRFLVALYTSRSRRFH*C*SGTLINQEWVLTAAN*C*DRKNIRIKLGMHSKN
AUS82526.1	IIGGEE*C*NINEHRFLVALYTFRSKRFH*C*SGTLINQEWVLTAAH*C*DRKNIRIKLGMHSTN
	100
rBamSP_1	VTNEDEQTRVPKEKFF*C*LSSKTYTKWDKDIMLIRLKRPVNDSPHIAPLSLPSNPPSVGS
Q7T229.1	VTNEDEQTRVPKEKFF*C*LSSKTYTKWDKDIMLIRLKRPVNDSPHIAPISLPSSPPSVGS
AUS82526.1	VTNEDAQTRVPKEKFF*C*LSSKTYTKWDKDIMLIRLKRPVNNSAHIATLSLPSNPPSLGS

rBamSP_1	V*C*RIMGWGTISPTKVSYPDVPH*C*ANINLLDYEV*C*RTAHGGLPATSRTL*C*AGILEGGKDS
Q7T229.1	V*C*RIMGWGTISPTKVSYPDVPH*C*ANINLLDYEV*C*RAAHGGLPATSRTL*C*AGILEGGKDS
AUS82526.1	V*C*RIMGWGTISATKETYPDVPH*C*ANINILDYEV*C*RAAHGGLPATSRTL*C*AGILKGGKDS
	197
rBamSP_1	*C*QGDSGGPLI*C*NGQFQGILSWGVHP*C*GQRLKPGVYTKVFDYTEWIRSIIAGNTDVT*C*PP
Q7T229.1	*C*QGDSGGPLI*C*NGQFQGILSWGVHP*C*GQRLKPGVYTKVSDYTEWIRSIIAGNTDVT*C*PP
AUS82526.1	*C*KGDSGGPLI*C*NGEIQGIVSWGAHP*C*GQSLKPGVYTKVFDYTEWIQSIIAGNTDAT*C*PP

^1^ Q7T229.1 and AUS82526.1 are serine-proteases from *Bothrops jararacussu* and *Crotalus mitchellii*, with 96.6% and 89.0% of identity to rBamSP_1, respectively. The residues in bold in positions His56, Asp100 and Ser197 represent the active site of serine-proteases.

**Table 2 toxins-11-00702-t002:** Amino acids residues of rBamMP_1 and alignment to similar viper enzymes.

rBamMP_1	Sequence ^1^
rBamMP_1	EQQRYNPYKYVEF*C*IVVDQGTVTKNNGDLDKIKTRIYELVNTVNEIYRYMYIHVALV*C*L
P30431.1	EQQRYDPYKYIEFFVVVDQGTVTKNNGDLDKIKARMYELANIVNEIFRYLYMHVALVGL
ALB00542.1	EQQKYNPFRYIEFLLVVDQGMVTKNNGDLDKIKARMYELANIVNEIFRYLYMHAALVGL

rBamMP_1	ETWSNGDKITVKPDVDYTWKSFAEWRKTVLLTRKNHDNAQLLTAIDFSGPTIGYAYIAT
P30431.1	EIWSNGDKITVKPDVDYTLNSFAEWRKTDLLTRKKHDNAQLLTAIDFNGPTIGYAYIGS
ALB00542.1	EIWSNGDKITVKPDVDYTLNSFAEWRKTDLLTRKKHDNAQLLTAIDFNGPTIGYAYIGS
	145-----------------156
rBamMP_1	M*C*DPKSSVGIVQDFSPINLLVAVTMA**HE**M**GH**NL**G**IH**HD**RGS*C*S*C*GGYP*C*IMGPVISNEP
P30431.1	M*C*HPKRSVGIVQDYSPINLVVAVIMA**HE**M**GH**NL**G**IH**HD**TGS*C*S*C*GDYPCIMGPTISNEP
ALB00542.1	M*C*HPKRSVAIVQDYSPINLVMAVIMA**HE**M**GH**NM**G**IH**HD**TGS*C*S*C*GDYPCIMGPTISNEP

rBamMP_1	SKFFSN*C*SYIQ*C*WDFIMNHNPE*C*IVNEPLGTDIVSPPV*C*GNELL
P30431.1	SKFFSN*C*SYIQ*C*WDFIMNHNPE*C*IINEPLGTDIISPPV*C*GNELL
ALB00542.1	SKFFSN*C*SYIQ*C*WDFIMNHNPE*C*IINEPLGPDIVSPPV*C*GNELL

^1^ P30431.1and ALB00542.1are metalloproteases from *Bothrops jararaca* and *Bothrops atrox,* with 86.4% and 83.2% of identity to rBamMP_1, respectively. The residues in bold positions 145-156, **HE**M**GH**NL**G**IH**HD,** represent the zinc binding motif of metalloproteases.

**Table 3 toxins-11-00702-t003:** Titers of antibodies from rabbits against the recombinant enzymes and *Bothrops* venoms.

Protein	Titers	CI ^1^
rBamPLA_2__1	17,148	15,705–18,723
rBamSP_1	10,673	9063–12,568
rBamMP_1	2653	2040–3451
*B. ammodytoides*	543	475–620
*B. jararacussu*	120	104–138
*B. diporus*	90	81–100
*B. moojeni*	624	572–679
*B. alternatus*	159	138–182
*B. asper*	85	80–91

^1^ Confidence intervals (95%).

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
