# Peer review of "Immunogenic Properties of Recombinant Enzymes from Bothrops ammodytoides towards the Generation of Neutralizing Antibodies against Its Own Venom"

_toxins, 2019, doi:10.3390/toxins11120702_

Round 1

Reviewer 1 Report

Please correct the following typos:

line 26: immunogens not inmunogens

line 48: species not specie

line 85: in vitro should be Italic

line 172: to not too

line 203: (metalloprotease, serine-protease and phospholipase) not (metalloproteases, serine-proteases and phospholipases)

line 225: instantaneously not nstantaneously

Please specify the methodology section:

line 170: in the description of  figure 6 is 37°C while in the methodology section room temperature

Figure 7. No description for RP-HPLC methodology in the method section.

line 283: Please specify how the samples were concentrated

In the method section, there is no description of protein identification, and the results are practically only in supplementary. Please link the added methodology with the caption of Figure 7.

Please specify the signatures of the figures:

Figure S3 - it is necessary to have an exact description of what is a spectrum, what is a chromatogram, why there are two sections for each sample

Figure 1 - the current description is incomprehensible and non-exhaustive

Figure 7 - Insufficient description of section c.

Other comments:

Lines 123-130 please add a clear link between these results (and the figure 4 and table 3) and the appropriate section in the methods

The results of the zymography are inconclusive. What are the dark bands in the upper regions of the gel? If these are whole IgG molecules or IgG molecules in complexes with target proteins, why did they not break down during denaturing electrophoresis? The description of the methodology shows that the conditions at the time of separation were denaturing and reducing, so please explain the presented results.

Author Response

Reviewer #1.

Please correct the following typos:

line 26: immunogens not inmunogens

Answer: it was corrected

line 48: species not specie

Answer: it was corrected

line 85: in vitro should be Italic

Answer: it was corrected

line 172: to not too

Answer: it was corrected

line 203: (metalloprotease, serine-protease and phospholipase) not (metalloproteases, serine-proteases and phospholipases)

Answer: it was corrected

line 225: instantaneously not nstantaneously

Answer: it was corrected

Please specify the methodology section:

line 170: in the description of  figure 6 is 37°C while in the methodology section room temperature.

Answer: overnight at room temperature

Figure 7. No description for RP-HPLC methodology in the method section.

Answer: it was included in the methodology at the Immunoaffinity chromatography section, the following: Once the venom was eluted from the affinity chromatography column, it was separated using an analytic C8 reverse-phase column (5C8MS, 4.6x250 mm, Zorbax Agilent) using a linear gradient with 0.1% trifluoroacetic acid, TFA, in water, as solvent A, and 0.1% TFA in acetonitrile as solvent B. The linear gradient was run from 0 to 60% solvent B, during 60 min at 1 mL/min, the peptide fragments were detected at 280 nm, collected and immediately lyophilized.

line 283: Please specify how the samples were concentrated

Answer: “were then lyophilized, and after, concentrated to 500 pmol/5 μL”

In the method section, there is no description of protein identification, and the results are practically only in supplementary. Please link the added methodology with the caption of Figure 7.

Answer: We include the description for protein identification, and now it is linked to the methodology, “For protein identification, the samples were previously reduced with dithiothreitol (Sigma-Aldrich; St Louis, MO, USA), alkylated with iodoacetamide (Sigma-Aldrich) and digested "in solution" with Trypsin (Promega Sequencing Grade Modified Trypsin; Madison, WI, USA). The peptides produced by enzymatic cleavage were desalted with Zip-Tip C18 (Millipore; Billerica, MA, USA) and applied to an LC-MS system composed of an EASY-nLC II nanoflow pump (Thermo-Fisher Co; San Jose, CA, USA) coupled to a LTQ-Orbitrap Velos MS (Thermo-Fisher Co, San Jose, CA, USA) with a nano-electrospray (ESI) ionization source. The total ion scanning (Full Scan) was performed on the Orbitrap analyzer with a resolution power of mass (RP Power; RP = m / FWHM) of 60,000. Peptide fragmentation was performed using the methods of CID (Collision-Induced Dissociation) and HCD (High-energy Collision Dissociation). All spectra were acquired in positive detection mode. The execution and capture of the fragmentation data were performed depending on the total ion scan according to the pre-determined charges (only ions with a z2+, z3+ and z4+ charge were fragmented) with an isolation width of 2.0 (m/z), normalized collision energy of 35 arbitrary units, Q activation of 0.250, activation time of 10 ms and maximum injection time of 10 ms per micro-scan. Protein identification was performed with the spectrometric data in raw format in the Proteome Discoverer program 1.4.1.14 (Thermo-Fisher Co., San Jose, CA, USA) through the Sequest HT search engine. For identity search, the bothrops. fasta (UniProt) protein database was used. An FDR-False Discovery Rate (Minimum) of 0.01 and FDR 0.05 (Maximum) was applied in addition to the inverted database (Decoy database) as a tool of the “Percolator” validation program. The maximum tolerance of molecular mass difference of the precursor ion when compared to the theoretical versus experimental values, ​​(precursor mass tolerance) was 20 ppm and the tolerance for the fragments obtained by dissociation of the precursor ion (fragment mass tolerance) was 0.6 Da. For the automatic search the following modifications were established: carbamido-methylation constants of cysteines (C) and variables, oxidation of methionines (M) and deamination of asparagine (N) and glutamine (Q).”

Please specify the signatures of the figures:

Figure S3 - it is necessary to have an exact description of what is a spectrum, what is a chromatogram, why there are two sections for each sample.

Answer: We have now included a description for Figure S3 “Figure S3: Mass spectrum of the recombinant proteins. A) rBamSP_1, the top spectrum comprises the raw data showing the total ion current (time vs intensity), the bottom spectrum is the deconvoluted spectrum on the true mass scale after Xcalibur Windows NT PC data system processing. B) rBamMP_1, the top spectrum comprises the raw data showing the total ion current according to the charge ionization, the bottom spectrum is the deconvoluted spectrum also processed by Xcalibur Data Acquisition and Interpretation Software (see Materials and Methods).”

Figure 1 - the current description is incomprehensible and non-exhaustive.

Answer: we include a description in the legend of figure 1 “The chromatographic separations of either rBamSP_1 or rBamMP_1 were from 400 mM imidazol fractions eluted from affinity chromatography. The recombinant proteins were separated using an analytic C4 reverse-phase column (Vydac 214 TP 4.6x250mm, USA) using 0.1% trifluoroacetic acid, TFA, in water, as solvent A, and 0.1% TFA in acetonitrile as solvent B. The gradient was run from 10 to 60% solvent B, during 50 min at 1 mL/min, the peptide fragments were detected at 230 nm. The HPLC fractions collected were analyzed using mass spectrometry.”

Figure 7 - Insufficient description of section c.

Answer: We include the description for protein identification of caption in section 5.12. Immunoaffinity chromatography.

Other comments:

Lines 123-130 please add a clear link between these results (and the figure 4 and table 3) and the appropriate section in the methods

Answer: Such results and the figure 4 and table 3 are now re-phrased.

The results of the zymography are inconclusive. What are the dark bands in the upper regions of the gel?

Answer: the dark bands are presumably complex of IgGs with some venom proteins.

If these are whole IgG molecules or IgG molecules in complexes with target proteins, why did they not break down during denaturing electrophoresis? The description of the methodology shows that the conditions at the time of separation were denaturing and reducing, so please explain the presented results.

Answer: probably, they do not break because we did not use denaturing conditions, only 12% SDS for the running gel and 5% Triton X-100 for removing the SDS. There was not boiling step or use of 2-mercaptoethanol; so, it may be not sufficiently strong to break such complexes. As a note, we did not use strong denaturing conditions because the venom proteases are needed for showing activity within the gelatin gel.

Reviewer 2 Report

Reviewer Dabor Resiere:

General Comments:

This is a very interesting observation, probably very important for  the production of antidotes or antivenoms

This article is a descriptive analysis revealing Immunogenic properties of recombinant enzymes from Bothrops ammodytoides towards the generation of neutralizing antibodies against its own venom

The descriptive and statistical analysis are excellent; the methodology as well. We know that horse-derived serum is an efficient therapy against snake venom, it is associated also with a high cost and side effects. Therefore, developing more cost-effective alternative treatment option is highly envisaged. Antibodies would have great potential for the development of therapeutic treatments against snake envenomation in the future. Up to date, there are some fully integrated manufacturer of polyclonal therapeutics, targeting infectious disease, snakebite events and others. Recombinant protein production is a valuable feature of this strategy. recombinant proteins uses to treat snake envenomation

The authors  provided informations  about the recombinant proteins were named rBamSP_1 and rBamMP_1 and  their composition.

This paper can be published after minor revision

Author Response

Reviewer #2.

General Comments:

This is a very interesting observation, probably very important for the production of antidotes or antivenoms. This article is a descriptive analysis revealing Immunogenic properties of recombinant enzymes from Bothrops ammodytoides towards the generation of neutralizing antibodies against its own venom.

The descriptive and statistical analysis are excellent; the methodology as well. We know that horse-derived serum is an efficient therapy against snake venom, it is associated also with a high cost and side effects. Therefore, developing more cost-effective alternative treatment option is highly envisaged. Antibodies would have great potential for the development of therapeutic treatments against snake envenomation in the future. Up to date, there are some fully integrated manufacturer of polyclonal therapeutics, targeting infectious disease, snakebite events and others. Recombinant protein production is a valuable feature of this strategy. recombinant proteins uses to treat snake envenomation

The authors provided information about the recombinant proteins were named rBamSP_1 and rBamMP_1 and their composition.

This paper can be published after minor revision

Answer: Thank you for your positive comments. Indeed, one aim of this work is to provide quality immunogens to improve the potency of viperid antivenoms. The manuscript was revised and several typos were corrected.